# Evaluating the Impact of a Virtual Health Coaching Lifestyle Program on Weight Loss after Sleeve Gastrectomy: A Prospective Study

**DOI:** 10.3390/healthcare12131256

**Published:** 2024-06-24

**Authors:** Kristina Strauss, Rachel Sauls, Michelle K. Alencar, Kelly E. Johnson

**Affiliations:** 1Department of Kinesiology, Coastal Carolina University, Conway, SC 29528, USA; kstrauss@coastal.edu; 2Department of Public Health, University of South Florida, Tampa, FL 33613, USA; rsauls@usf.edu; 3InHealth Medical Services, Inc., Los Angeles, CA 90067, USA; malencar@inhealthonline.com; 4Department of Kinesiology, California State University Long Beach, Long Beach, CA 90840, USA

**Keywords:** bariatric surgery, health and wellness coaching, obesity, telehealth

## Abstract

Bariatric surgery (BS) is a leading treatment for obesity; however, adverse side effects (e.g., pain and infection) can deter patients or affect weight maintenance. This study investigates how a post-operative virtual health coaching lifestyle program, monitoring virtual weekly goal progress made by patients, affects weight loss after BS, specifically sleeve gastrectomy. Patients recruited for this 6-month study were classified with a BMI > 30 kg/m^2^ 90 days post-operatively. Patients were prescribed lifestyle support delivered by certified health and wellness coaches (InHealth Lifestyle Therapeutics™). Demographic variables (e.g., age, weight, height, and gender) were obtained and compared according to initial, 3-, 6-month, and current weight through repeated measures ANOVA and post hoc comparison. Thirty-eight adult participants were included, with a mean age of 52 years ± 12.9 and with a majority (n = 35; 97%) of them being female. There were significant differences in weight reported across all five time points (*p* < 0.05), with the greatest weight difference seen between the initial (250.3 ± 45.5 lbs.) and final time points (226.7± 40.4 lbs.). This study suggests post-operative virtual health coaching can enhance weight loss outcomes after sleeve gastrectomy. Further research is needed to assess the long-term effects and cost-effectiveness of such a form of coaching for bariatric surgery patients.

## 1. Introduction

As of 2013, the American Medical Association has recognized obesity as a disease, advocating for treatment and prevention efforts [1]. Obesity poses severe physical and psychological problems for patients and has economic implications for the US healthcare system [2,3]. Obesity is classified as having a body mass index of 30 kg/m^2^ or higher [4]. Although many treatment methods exist for obesity, bariatric surgery (BS) is often pursued when patients have exhausted all other options. BS assists patients with obesity by altering the digestive system to aid in weight loss by lowering food consumption [5]. A specific form of BS, sleeve gastrectomy, reduces appetite while taking out a portion of the stomach [6]. Sleeve gastrectomy (SG) is increasingly preferred for reducing obesity because it achieves substantial and sustained weight loss, relatively lower complication rates, and shorter recovery times than other bariatric procedures [7]. A recent sleeve gastrectomy review explains that the effects of this procedure may include reduced blood pressure, cardiovascular disease, type 2 diabetes, and stroke [8]. In addition, this procedure has created a higher survival rate for adults with morbid obesity [8].

SG is an effective treatment to fight obesity, but the ability to sustain weight loss can be challenging, and weight regain is a common occurrence post-surgery [6,9]. Following post-operation, patients meet with a bariatric dietician, are specifically counseled on post-operative meal stages, and receive periodic counseling when transitioning to an entire food nutrition plan. However, only 40% of patients return for their first four annual follow-up visits [10,11]. Despite SG, patients still report poor dietary intake and sedentary lifestyles [9]. Furthermore, even after undergoing sleeve gastrectomy, many patients continue to struggle with poor dietary intake and maintain sedentary lifestyles [12], which can impede long-term success and overall health improvement [13]. This underscores the need for enhanced post-operative support and interventions to help patients achieve and maintain their weight loss goals.

In contrast, social support, post-operative follow-up visits, and self-monitoring of weight have been associated with weight maintenance [14,15,16]. One strategy to enhance post-operation BS program adherence and weight loss is a virtual health coaching lifestyle program utilizing certified health coaches trained in lifestyle modifications necessary after BS [17]. The growing body of evidence supporting the effectiveness of virtual health coaching in other medical contexts further strengthens the rationale for this study [18]. Virtual health coaching has shown significant benefits in managing chronic conditions such as diabetes, hypertension, and cardiovascular diseases by providing personalized, continuous support that improves patient engagement and adherence to treatment plans [19,20]. These interventions have led to better clinical outcomes, enhanced patient satisfaction, and increased self-efficacy in managing health behaviors [21]. Furthermore, telehealth can help facilitate and increase the follow-up rate, be cost-effective, and reduce patient burden [22]. Although virtual health coaching lifestyle programs have been utilized between clients and practitioners in clinical subspecialties (i.e., heart disease, diabetes, and nutritional care) [17,23], there is a lack of literature regarding its use, specifically in BS settings.

Virtual health coaching presents a promising solution for addressing the challenges faced by bariatric surgery patients [24]. By offering convenient, ongoing support, virtual health coaching can help improve dietary habits, encourage physical activity, and provide the necessary motivation for patients to adhere to follow-up visits and self-monitoring practices [25,26]. This study investigates the effectiveness of a post-operative virtual health coaching lifestyle program in enhancing weight loss outcomes after sleeve gastrectomy. By leveraging the successes of virtual health coaching in other medical contexts, this study seeks to provide a robust rationale for its application in bariatric care, ultimately contributing to improved long-term weight maintenance and overall patient well-being.

## 2. Materials and Methods

### 2.1. Participants

Patients were recruited for this 6-month study using the following inclusion criteria: adult (>18 years) males or females who were classified as having a BMI > 30 kg/m^2^, had undergone a sleeve gastrectomy, were at 90 days post-operation, and had no life-threatening complications related to the surgery or other conditions (e.g., chronic conditions, infections, or hospital re-admissions). The study was conducted in accordance with the guidelines of the Declaration of Helsinki and approved by the Institutional Review Board of Coastal Carolina University (IRB #2021.96, approval date 3 March 2022). Following sleeve gastrectomy, patients were prescribed by their bariatric surgeon the InHealth Lifestyle Therapeutics™ program, a telehealth-based virtual program delivered by certified health and wellness coaches. 

### 2.2. Virtual Health Coaching Program

The InHealth Lifestyle Therapeutics™ (IHLT) [27] program was broken down into three phases: (1) weekly videoconference-based visits with a nationally certified health coach for the first 12 weeks, (2) bi-weekly visits with the health coach for the next 12 weeks, and (3) monthly visits for the next six months. Health coaching was based on weekly progress made by patients toward goals.

Participants initially completed a 15–20 min introductory visit via telehealth using the InHealth ^®^ app. During this visit, the patient met with a certified health coach, where they discussed and reviewed all relevant medical history while also giving an overview of the InHealth Lifestyle Therapeutic™ program. Next, an initial visit was scheduled via videoconferencing, which included the patient working with the health coach to set a vision, 30-day and 6-month SMART goals, and action steps. Following the introduction and initial visits, the following visits were scheduled. Each follow-up visit lasted between 25 and 30 min. During the follow-up visits, action steps were reset at every visit, and goals were revisited monthly and at six months. Each participant received unique care for their needs, and the summary of the visits can be found in Table 1. Coaches checked in with patients periodically through the InHealthh ^®^ app to ensure the maintenance of goals.

Patients followed the surgeon’s post-operation nutrition plan, which included four phases: (1) 1–3 days of clear liquids, (2) 4–14 days of full liquids, (3) 14–21 days of pureed foods, (4) soft foods, and (5) a regular diet. The surgeons recommended 48–64 ounces of water daily and 60 g of protein, avoiding starches such as bread, chips, pasta, candy, and rice but focusing on complex carbs such as vegetables and legumes. Patients were encouraged to take two chewable vitamins with iron and calcium daily. The surgeon also recommended that patients walk daily and, with time, include strength-based exercise to help with muscle mass maintenance and weight loss. To ensure the patients followed the surgical plan, constant communication was maintained between coaches and patients, with weekly, biweekly, and monthly check-ins on adherence and weight.

### 2.3. Data Variables

Demographic data (e.g., age, race, and gender) were extracted from medical charts filled out by the coaching team at InHealth and were self-reported by participants during the first meeting. Weight measurements (e.g., weight and height) were taken at each coaching session and self-reported by the patient at all five time points of the project. Body mass index was calculated based on weight and height collected from the patients. All data were kept on a secure database belonging to the company.

### 2.4. Statistical Analysis

Body weight changes across five different time points, including initial, 1-month, 3-month, 6-month, and current weight, were analyzed using repeated measures ANOVA. Post hoc analysis pairwise comparisons were then performed if the results were significant. All data were analyzed using SPSS, V.27 Statistics for Windows, Version 25.0. Armonk, NY, USA: IBM Corp.), with data displayed as average  ±  SD and the significance set to *p*  <  0.05.

## 3. Results

In total, n = 38 adult participants were included in this population; however, 2 patients were excluded from statistical analysis due to missing data. The mean age was 52 years ± 12.9, with n = 35 females (97%). Height and initial weight averaged 65 in ± 2.6 in. and 271 lbs. ± 53.1 lbs., respectively. Changes in BMI ranged between −26.2 kg/m^2^ and 2.65 kg/m^2^ (mean −7.4 kg/m^2^ ± 6.7 kg/m^2^); the mean weights reported show a downward trend, as seen in Table 2 and Figure 1. A post hoc pairwise comparison using the Bonferroni correction showed a significant weight decrease between all time points, as seen in Figure 1 (*p* < 0.05). However, the difference between weight reported at six months and current weight was insignificant (*p* = 0.169).

Significant differences were observed for weight reported across all five time points, including initial, 1-month, 3-month, 6-month, and current weight (*p* < 0.05). Repeated measures ANOVA reported significant weight loss findings across five time points (*p* < 0.001).

## 4. Discussion

This study investigated the impact of a post-operative virtual health coaching lifestyle program on weight loss after sleeve gastrectomy. The results demonstrate significant weight loss among participants at multiple time points following the surgery, indicating the effectiveness of the virtual health coaching program in supporting weight management.

The repeated measures ANOVA revealed a significant decrease in weight across all five time points (initial, one month, three months, six months, and current; *p* < 0.001). Post hoc pairwise comparisons using the Bonferroni correction demonstrated a significant decrease in weight between all time points (*p* < 0.05), except between the 6-month and current weight (*p* = 0.169). This finding suggests that the weight loss achieved during the first six months was maintained at the current weight, indicating the potential for sustained weight management with the virtual health coaching program. This is similar to current findings indicating that an online platform can effectively maintain weight loss and management [28].

The results of this study align with those of previous research demonstrating the effectiveness of health coaching in promoting weight loss and lifestyle modifications. The virtual delivery of the coaching program offers several advantages, including increased accessibility, reduced patient burden, and cost-effectiveness [29]. By utilizing telehealth technology, patients can receive ongoing support and guidance from health coaches without needing in-person visits [30]. Compared with other post-operative support programs, particularly non-virtual ones, the virtual health coaching program provides continuous and flexible support that can adapt to the patient’s schedule, reducing barriers to consistent follow-up [31,32]. Traditional in-person programs, while effective, often face challenges such as limited accessibility and higher costs, which can hinder patient adherence [33]. This study adds to the growing body of evidence supporting virtual health interventions, showing that they can be as effective, if not more so, than their in-person counterparts.

The results of this study are consistent with those of previous research that has demonstrated the effectiveness of health coaching in promoting weight loss and lifestyle modifications [17,23]. The virtual delivery of the coaching program offers several advantages, including increased accessibility, reduced patient burden, and cost-effectiveness [34,35]. Similar to findings from previous research, reduced healthcare burden through telehealth improved adherence and outcomes, especially among post-operative patients [32]. The virtual delivery of health coaching represents a promising approach to improving post-operative care for bariatric surgery patients [36]. Thus, integrating virtual health coaching into routine practice can enhance patient engagement, optimize treatment outcomes, and ultimately improve long-term health and well-being.

The results of this study hold significant implications for the field of bariatric surgery and weight management. The high adherence rate observed in the virtual health coaching program underscores its potential to enhance post-operative care and contribute to long-term weight loss success [37]. Through personalized support, guidance, and accountability, health coaches play a crucial role in assisting patients as they navigate the complexities of post-operative lifestyle adjustments [38]. This includes facilitating dietary modifications tailored to individual needs and promoting regular physical activity to support ongoing weight management efforts [38]. By fostering a supportive environment and offering tailored interventions, virtual health coaching can empower patients to adopt sustainable lifestyle changes, ultimately improving their overall health and long-term well-being [31].

However, it is essential to acknowledge some limitations of this study. Firstly, the sample size was relatively small, which may limit the generalizability of the findings. Future studies with more extensive and diverse populations are needed to validate the results. Additionally, most of the study population included women, and this can reduce generalizability to the male population. Future projects should prioritize equal recruitment between male and female participants. Secondly, the study did not include a control group, making it difficult to determine the specific contribution of the virtual health coaching program to the observed weight loss. Comparative studies that include a control group receiving standard care would provide a more comprehensive understanding of the program’s effectiveness. Additionally, this study utilized self-reported anthropometric data (e.g., weight and height), which possible discrepancies or response bias could be attributed to. Future studies can be adjusted by providing scales or measuring anthropometrics in person with research staff. 

## 5. Conclusions

Our study underscores the effectiveness of a post-operative virtual health coaching lifestyle program in supporting weight loss and maintenance among individuals who have undergone sleeve gastrectomy. The findings emphasize the crucial role of ongoing support and guidance in achieving sustained weight management outcomes. The virtual delivery of such programs presents a convenient and accessible approach that can be seamlessly integrated into routine clinical practice. Moving forward, further research is warranted to delve into the long-term effects and cost-effectiveness of virtual health coaching interventions within bariatric surgery. Additionally, we recommend the widespread adoption of virtual health coaching programs as a valuable tool in enhancing post-operative care and optimizing long-term weight loss success for bariatric surgery patients.

## Figures and Tables

**Figure 1 healthcare-12-01256-f001:**
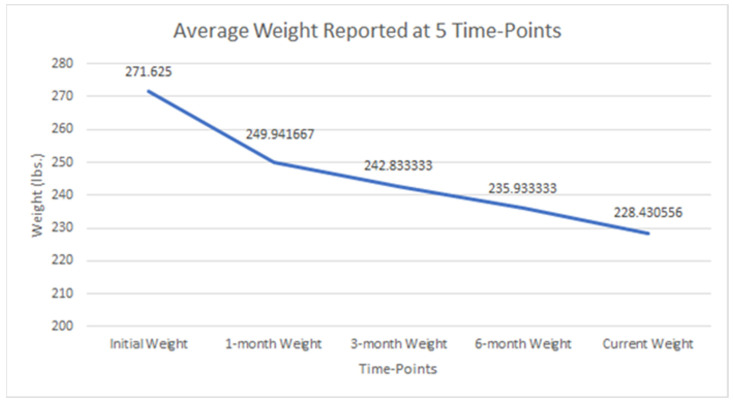
Mean weight reported across five time points (*p* < 0.001).

**Table 1 healthcare-12-01256-t001:** Health and wellness coaching session plan.

Weekly sessions (months 1–3)	Build and maintain rapport Create progress toward 12-month wellness vision Motivational interviewing to support generative moments Discuss nutritional and physical activity goals and progress
Biweekly sessions (months 4–6)	Continue maintaining rapport Explore strengths and values to health behavior and change Troubleshoot barriers and highlight strengths Bi-directional feedback with MD
Monthly sessions (months 6–12)	Continue maintaining rapport Schedule subsequent health coaching visits Goal setting—monthly Continue reviewing strengths and weaknesses

**Table 2 healthcare-12-01256-t002:** Participant characteristics.

	N = 38	*p* Value
**Age:** Mean ± SD [range], years	52.3 ± 12.9 [24.8–75.1]	
**Initial BMI:** Mean ± SD [range]	44.5 ± 8.6 [27.9–70.6]	<0.001 *
**Current BMI:** Mean ± SD [range]	37.1 ± 6.6 [25.7–51.1]	
**Gender:** n (%) Female	35 (97.0)	
**Height:** Mean ± SD [range], inches	65.6 ± 2.6 [59–71.2]	
**Initial Weight:** Mean ± SD [range], kg.	123.3 ± 24.1 [77.1–191.9]	<0.001 *
**1-month Weight:** Mean ± SD [range], kg.	113.5 ± 20.6 [75.8.0–163.3]	<0.001 *
**3-month Weight:** Mean ± SD [range], kg.	110.1 ± 19.5 [73.5–155.6]	<0.001 *
**6-month Weight:** Mean ± SD [range], kg.	106.8 ± 18.8 [72.2–149.2]	<0.001 *
**Current Weight:** Mean ± SD [range], kg.	102.8± 18.3 [71.2–149.2]	<0.001 *

Note: BMI = body mass index; SD = standard deviation; kg = kilograms; * *p* < 0.05.

## Data Availability

The data presented in this study are available upon request from the corresponding author. The data are not publicly available due to privacy and ethical considerations of patients’ ongoing coaching sessions with the company.

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
