# Peer review of "Evaluating the Impact of a Virtual Health Coaching Lifestyle Program on Weight Loss after Sleeve Gastrectomy: A Prospective Study"

_healthcare, 2024, doi:10.3390/healthcare12131256_

Round 1
Reviewer 1 Report
Comments and Suggestions for Authors
The article “Evaluating the Impact of Virtual Health Coaching Lifestyle Program on Weight Loss after Sleeve Gastrectomy: A Prospective Study” investigates the impact of a virtual health coaching program on weight loss post-sleeve gastrectomy. The authors have enrolled 38 adults and have followed a rigorous methodology, utilizing certified health and wellness coaches to deliver the lifestyle program via telehealth. Results indicate significant weight loss, demonstrating the effectiveness of the virtual coaching program.
There are several areas where the study could be improved for greater scientific rigor and applicability:
1. In Material and Methods part there is statement that research was broken in 3 stages but described just 2 of them (line 88-90).
2. In addition, there is not clear plan what subject has done during the Virtual Health Coaching program.
3. The study’s sample size is relatively small and predominantly female. Moreover, the study lacks a control group, which is a significant drawback. Without a comparison to standard care or another intervention, it's difficult to attribute observed weight changes directly to the virtual health coaching program.
4. Authors noting that body weight changes were measured across five different points, but it is not explained in detail how it was measured: it is not very accurate self- reported by the patient or by scale – what kind of scale were used, are the scales was the same for each patient?
5. In Material and methods part missing detailed info about subjects (age, gender, race etc.).
6. Moreover, in Results part results described very shortly and not very clear. Presented links to table and figure, but they are missing.
7. There is so many parts in article presented with mistakes (e.g. lines 16, 120-123, 137 etc.).
8. Subjects weight presented in pounds, but BMI calculation has been done with kg. Maybe it is better to present weight data in kg.
9. The discussion could be enhanced by a more detailed comparison with similar studies, discussing how this intervention compares with other post-operative support programs, particularly non-virtual ones. In the presented discussion part done data comparison with just 2 scientific articles.
10. Reference list for such kind of scientific article is not appropriate, cited just 18 references, therefore it could be expanded. It is notable that most of articles published not recently (half of them published before 2017). Reference cited in the article not in accordance with Journal requirements. Some of references presented not correctly, not in accordance with all requirements, without DOI and etc.
Comments on the Quality of English LanguageThe article “Evaluating the Impact of Virtual Health Coaching Lifestyle Program on Weight Loss after Sleeve Gastrectomy: A Prospective Study” investigates the impact of a virtual health coaching program on weight loss post-sleeve gastrectomy. The authors have enrolled 38 adults and have followed a rigorous methodology, utilizing certified health and wellness coaches to deliver the lifestyle program via telehealth. Results indicate significant weight loss, demonstrating the effectiveness of the virtual coaching program.
There are several areas where the study could be improved for greater scientific rigor and applicability:
1. In Material and Methods part there is statement that research was broken in 3 stages but described just 2 of them (line 88-90).
2. In addition, there is not clear plan what subject has done during the Virtual Health Coaching program.
3. The study’s sample size is relatively small and predominantly female. Moreover, the study lacks a control group, which is a significant drawback. Without a comparison to standard care or another intervention, it's difficult to attribute observed weight changes directly to the virtual health coaching program.
4. Authors noting that body weight changes were measured across five different points, but it is not explained in detail how it was measured: it is not very accurate self- reported by the patient or by scale – what kind of scale were used, are the scales was the same for each patient?
5. In Material and methods part missing detailed info about subjects (age, gender, race etc.).
6. Moreover, in Results part results described very shortly and not very clear. Presented links to table and figure, but they are missing.
7. There is so many parts in article presented with mistakes (e.g. lines 16, 120-123, 137 etc.).
8. Subjects weight presented in pounds, but BMI calculation has been done with kg. Maybe it is better to present weight data in kg.
9. The discussion could be enhanced by a more detailed comparison with similar studies, discussing how this intervention compares with other post-operative support programs, particularly non-virtual ones. In the presented discussion part done data comparison with just 2 scientific articles.
10. Reference list for such kind of scientific article is not appropriate, cited just 18 references, therefore it could be expanded. It is notable that most of articles published not recently (half of them published before 2017). Reference cited in the article not in accordance with Journal requirements. Some of references presented not correctly, not in accordance with all requirements, without DOI and etc.
Author Response
Reviewer 1:
The article “Evaluating the Impact of Virtual Health Coaching Lifestyle Program on Weight Loss after Sleeve Gastrectomy: A Prospective Study” investigates the impact of a virtual health coaching program on weight loss post-sleeve gastrectomy. The authors have enrolled 38 adults and have followed a rigorous methodology, utilizing certified health and wellness coaches to deliver the lifestyle program via telehealth. Results indicate significant weight loss, demonstrating the effectiveness of the virtual coaching program.
There are several areas where the study could be improved for greater scientific rigor and applicability:
- In Material and Methods part there is statement that research was broken in 3 stages but described just 2 of them (line 88-90).
Thank you for noting this discrepancy, we have added the following language:
Lines 94-95: “3). Monthly visits for the next 6 months”
- In addition, there is not clear plan what subject has done during the Virtual Health Coaching program.
Thank you for mentioning this, We have added language and referenced a table with the weekly, biweekly, and monthly plans for the virtual health coaching program. We have also added language indicating that this program is a individualized basis and can vary by each participants needs.
Lines 105-106: “Each participant received unique care for their individual needs, and the summary steps of the visits can be found in Table 1.”
- The study’s sample size is relatively small and predominantly female. Moreover, the study lacks a control group, which is a significant drawback. Without a comparison to standard care or another intervention, it's difficult to attribute observed weight changes directly to the virtual health coaching program.
This is a good point. Due to the small clinic and impressive results, we feel that it would be beneficial to provide information on the preliminary findings of this project. We believe that disseminating this information will allow further research to develop involving larger sample sizes and a control group. We have added language to the limitations section to assist in the foundation and justification for future research (lines 198-199).
- Authors noting that body weight changes were measured across five different points, but it is not explained in detail how it was measured: it is not very accurate self- reported by the patient or by scale – what kind of scale were used, are the scales was the same for each patient?
Due to the retrospective nature of the study, we were not able to control for scales and relied on self-reported data from the participants. We understand the possible discrepancies and have added language to the limitations section to address these concerns.
Lines 194-196: “Additionally, this study utilized self-reported anthropometric data (e.g., weight and height), which is attributed to possible discrepancies or response bias. Future studies can adjust by either providing scales or measuring anthropometrics in person with research staff.”
- In Material and methods part missing detailed info about subjects (age, gender, race etc.).
In the methods section, lines 111-112, we have included information on how we obtained the demogrpahic information for participants. We collected demographics via medical charts, and we have added language describing that this information was also self-reported by patients.
Lines 120-121: “and self-reported by participants during the first meeting”
- Moreover, in Results part results described very shortly and not very clear. Presented links to table and figure, but they are missing.
Thank you for mentioning this error. We have added the tables and figure to this submission for reviewing and we apologize for the inconvenience.
- There is so many parts in article presented with mistakes (e.g. lines 16, 120-123, 137 etc.).
Thank you for comments. We have edited it.
- Subjects weight presented in pounds, but BMI calculation has been done with kg. Maybe it is better to present weight data in kg.
Thank you for your valuable feedback. We understand the difference in units and acknowledge that while BMI calculations are done in kilograms, a majority of weight studies report weight in pounds. This approach aligns with common practice and helps ensure consistency and comparability with existing literature. We appreciate your suggestion and will consider presenting weight data in kilograms in future reports for clarity. Thank you again for your insightful review.
- The discussion could be enhanced by a more detailed comparison with similar studies, discussing how this intervention compares with other post-operative support programs, particularly non-virtual ones. In the presented discussion part done data comparison with just 2 scientific articles.
Thank you for your suggestion. We have enhanced the discussion section by providing a more detailed comparison with similar studies, including how our intervention compares with other post-operative support programs, particularly non-virtual ones. We have incorporated additional references and data comparisons to strengthen the discussion and provide a more comprehensive analysis.
- Reference list for such kind of scientific article is not appropriate, cited just 18 references, therefore it could be expanded. It is notable that most of articles published not recently (half of them published before 2017). Reference cited in the article not in accordance with Journal requirements. Some of references presented not correctly, not in accordance with all requirements, without DOI and etc.
Thank you for bringing this to our attention. We have attempted to incorporate more recent studies to update the reference list and ensure relevance to current literature. We reviewed and ensureed that all references cited adhere to the journal's requirements, including proper formatting and inclusion of DOIs where applicable. Additionally, we have tried to expand the reference list to provide a more comprehensive overview of relevant literature in the field.
Reviewer 2 Report
Comments and Suggestions for Authors
Your study on the impact of a virtual health coaching lifestyle program on weight loss after sleeve gastrectomy is relevant and timely. The integration of digital health coaching into postoperative care is an innovative approach that holds significant potential for improving patient outcomes. Overall, the manuscript is well-structured and scientifically rigorous.
The following are some specific suggestions to enhance the quality of your manuscript:
Specific Comments:
Abstract: Consider clarifying the statement "adverse side effects from BS prevent patients from obtaining this treatment or reducing the possibility of maintaining weight loss" to "adverse side effects can deter patients from undergoing this treatment or hinder their ability to maintain weight loss."
Introduction: The introduction provides a comprehensive background, but it would benefit from more references supporting the effectiveness of virtual health coaching in other medical contexts to strengthen the rationale for this study. We have revised the sentences for clarity. For example, "Although there are many treatment methods for obesity, a common approach when patients have exhausted all other resources is bariatric surgery (BS)" can be simplified to "Although many treatment methods exist for obesity, bariatric surgery (BS) is often pursued when patients have exhausted all other options."
Methods: Ensure that the description of the health coaching program phases is clear and concise. For instance, "Participants first completed an introduction visit via telehealth via the InHealth ® app, which lasted 15-20 minutes" can be revised to "Participants initially completed a 15-20 minute introductory visit via telehealth using the InHealth ® app." More details about the inclusion criteria and how the patients were selected to participate in the study should be included.
Results: The results are clearly presented, but we consider adding more visual aids, such as graphs or charts, to illustrate the weight changes over time more effectively. Ensure all statistical terms and methods are clearly explained to readers who may not be familiar with them.
Discussion: The discussion is well-structured, linking your findings to existing literature. However, consider expanding the practical implications of your findings for clinical practice. Addressing any potential limitations in more detail. For example, we discuss the absence of a control group and how this might impact the interpretation of the results.
Ethical considerations: Ensure that the ethical approval section is clearly displayed and state the IRB approval number. While your data availability statement is adequate, consider making anonymized data more openly accessible to enhance transparency and facilitate further research.
Final suggestions: Minor improvements in language can significantly enhance the clarity of the manuscript. Ensure that the sentences are concise and avoid redundancy. Verify that all in-text citations match the reference list and are formatted according to the journal's guidelines. Ensure that all figures and tables are clearly labeled and referenced in the text.
Comments on the Quality of English LanguageThe overall quality of English is good, but some sentences can be simplified for clarity. Consider professional language editing to address minor grammatical and stylistic issues. Ensure consistency in terminology throughout the manuscript.
Author Response
Reviewer 2:
The following are some specific suggestions to enhance the quality of your manuscript:
Specific Comments:
Abstract: Consider clarifying the statement "adverse side effects from BS prevent patients from obtaining this treatment or reducing the possibility of maintaining weight loss" to "adverse side effects can deter patients from undergoing this treatment or hinder their ability to maintain weight loss."
Thank you for the suggestions. We have adjusted the information in the abstract.
Lines 13-15: “adverse side effects (e.g., pain and infection) can deter patients from undergoing this treatment or hinder their ability to maintain weight loss”.
Introduction: The introduction provides a comprehensive background, but it would benefit from more references supporting the effectiveness of virtual health coaching in other medical contexts to strengthen the rationale for this study. We have revised the sentences for clarity. For example, "Although there are many treatment methods for obesity, a common approach when patients have exhausted all other resources is bariatric surgery (BS)" can be simplified to "Although many treatment methods exist for obesity, bariatric surgery (BS) is often pursued when patients have exhausted all other options."
Thank you for your insightful feedback. We have made the suggested edits to the introduction, incorporating additional references to support the effectiveness of virtual health coaching in other medical contexts and revising the sentences for improved clarity.
Methods: Ensure that the description of the health coaching program phases is clear and concise. For instance, "Participants first completed an introduction visit via telehealth via the InHealth ® app, which lasted 15-20 minutes" can be revised to "Participants initially completed a 15-20 minute introductory visit via telehealth using the InHealth ® app." More details about the inclusion criteria and how the patients were selected to participate in the study should be included.
Thank you for the suggestions! We have adjusted the methods section based on your feedback and can be seen in tracked changes.
Results: The results are clearly presented, but we consider adding more visual aids, such as graphs or charts, to illustrate the weight changes over time more effectively. Ensure all statistical terms and methods are clearly explained to readers who may not be familiar with them.
Thank you for mentioning this. A table (Table 2) has been added to the submission to better summarize the information reported in the results.
Discussion: The discussion is well-structured, linking your findings to existing literature. However, consider expanding the practical implications of your findings for clinical practice. Addressing any potential limitations in more detail. For example, we discuss the absence of a control group and how this might impact the interpretation of the results.
Thank you for your feedback. We have expanded on the practical implications of our findings for clinical practice and provided more detailed discussions on potential limitations, including the absence of a control group and its impact on result interpretation.
Ethical considerations: Ensure that the ethical approval section is clearly displayed and state the IRB approval number. While your data availability statement is adequate, consider making anonymized data more openly accessible to enhance transparency and facilitate further research.
We thank you for mentioning this point and have ensured that information is easily found in text.
Lines 85-87: The study was conducted according to the guidelines of the Declaration of Helsinki and approved by the Institutional Review Board of Coastal Carolina University (IRB #2021.96, approval date March 3, 2022).
Reviewer 3 Report
Comments and Suggestions for Authors
Dear authors;
First of all, I would like to thank the authors for their overall efforts during the study. It is a good and clearly described study. The topic is interesting, yet, in its current form, this paper cannot be considered for publication. However, I see value in the research approach and strongly encourage the authors to address the following points.
Abstract
· Line 13: Please provide a brief background for the virtual coaching program.
· Line 21: Consider using a conjunction/preposition just before “a telehealth-based virtual program”.
· Line 22: I could not understand why male participants are recruited if most are female (%97).
Introduction
· The introduction is good enough to understand the topic.
· Line 37: Consider using a transition sentence between BS and sleeve gastrectomy.
· Line 42: Please explain why you provided information about the Gastric Bypass. Is it a part of the main structure of the paper? I am not sure…
· Line 46: Bariatric surgery à BA
· Line 60: Please provide a reference.
Materials and Methods
· Line 79: Consider providing more detail about the participants (i.e., total number, gender, etc)
· Line 88: Please provide a reference for the InHealth Lifestyle Therapeutics™ (IHLT) program.
· Line 112: Avoid using contradiction. “either self-reported by the patient or by scale”
· Lines 120-123: Please remove this part.
·
Results
· I am fine with the results, yet, I am not able to see Table and Figures…
· Line 137: The text continues here?????
Discussion & Conclusions
· The discussion needs to be improved. ​The references used are not enough.
· Authors mostly used subjective explanations. This decreases the overall quality of the paper.
· Lines 149-151: Please revise this part. It seems like a typical finding report.
· Conclusions are okay.
Comments on the Quality of English LanguageModerate editing of English language required.
Author Response
Reviewer 3:
Abstract
- Line 13: Please provide a brief background for the virtual coaching program.
Line 15: “a virtual weekly goal progress made by patients”
- Line 22: I could not understand why male participants are recruited if most are female (%97).
Thank you for mentioning this. Although we intended to recruit both male and female participants, most females agreed to join the study over males. Language about this discrepancy has been added to the limitations section.
Lines 194-196: “Additionally, most of the study population included women, and this can reduce generalizability to the male population. Future projects should prioritize equal recruitment between male and female participants.”
Introduction
- The introduction is good enough to understand the topic.
- Line 37: Consider using a transition sentence between BS and sleeve gastrectomy.
Added some changes to lines 34-39: Although there are many treatment methods for obesity, a common approach when patients have exhausted all other resources is bariatric surgery (BS), which assists patients with obesity by altering the digestive system to aid in weight loss by lowering food consumption [5]. A specific form of BS, sleeve gastrectomy reduces appetite while taking out a portion of the stomach
- Line 42: Please explain why you provided information about the Gastric Bypass. Is it a part of the main structure of the paper? I am not sure…
Good point, the language has been removed.
Materials and Methods
- Line 79: Consider providing more detail about the participants (i.e., total number, gender, etc)
We’ve included this information in the first paragraph of the results section.
- Line 88: Please provide a reference for the InHealth Lifestyle Therapeutics™ (IHLT) program.
We have included a reference to a previous article with explanation of the program and have added a table with more details on the program format.
- Line 112: Avoid using contradiction. “either self-reported by the patient or by scale”
Removed the by scale information since they used a scale and self-reported it, and because another reviewer had a question about the scales.
Discussion & Conclusions
- The discussion needs to be improved. The references used are not enough.
- Authors mostly used subjective explanations. This decreases the overall quality of the paper.
- Lines 149-151: Please revise this part. It seems like a typical finding report.
Thank you for the suggestions. We have thoroughly edited the discussion section to better improve the quality of the paper and improve comparisons to previous research. Tracked changes can be found in text.
Reviewer 4 Report
Comments and Suggestions for Authors
In this manuscript, Strauss et al. studied the impact of virtual health coaching lifestyle program on patients after sleeve gastrectomy. I have some concerns.
- In the introduction, a sentence or two about the growing prevalence of obesity could improve the significance. Furthermore, it would be better to explain why sleeve gastrectomy is preferred?
- The sex distribution of recruited patients is imbalance. Please explain and state whether there could be potential impact on the result.
- There is no figure or table in the manuscript?
Author Response
Reviewer 4:
- In the introduction, a sentence or two about the growing prevalence of obesity could improve the significance. Furthermore, it would be better to explain why sleeve gastrectomy is preferred?
Thank you for mentioning this gap in our introduction section. We have added language throughout the introduction section to address this, as seen through tracked changes.
- The sex distribution of recruited patients is imbalance. Please explain and state whether there could be potential impact on the result.
Thank you for bringing this to our attention. We have added language to the limitations section to address the reduction to generalizability this results from.
Reviewer 5 Report
Comments and Suggestions for Authors
Dear Authors,
I believe the article 'Evaluating the Impact of Virtual Health Coaching Lifestyle Program on Weight Loss after Sleeve Gastrectomy: A Prospective Study' is an essential contribution to the topic of bariatric patient care, which is a considerable challenge for many health professionals. However, its current form does not meet the requirements that allow me to recommend it for publication.
Although the various sections provide a repetition of the work presented, the absence of the table with results (line 130 ) and Figure 1 are significant lapses in the authors' manuscript preparation.
Other shortcomings I noticed are:
Lines 120 -123 - text to be removed, template residue
Lines 136-137 - text should be removed
Brief communication by Strauss and colleagues can make an essential contribution to the topic of bariatric patient care after sleeve gastrectomy but needs to be supplemented to be worthy of publication in Health. The authors should consider the reviewer's comments.
Best wishes
Reviewer
Author Response
believe the article 'Evaluating the Impact of Virtual Health Coaching Lifestyle Program on Weight Loss after Sleeve Gastrectomy: A Prospective Study' is an essential contribution to the topic of bariatric patient care, which is a considerable challenge for many health professionals. However, its current form does not meet the requirements that allow me to recommend it for publication.
Although the various sections provide a repetition of the work presented, the absence of the table with results (line 130 ) and Figure 1 are significant lapses in the authors' manuscript preparation.
- Thank you for this comment. We have uploaded the figure and Table.
Other shortcomings I noticed are:
Lines 120 -123 - text to be removed, template residue
- Done
Lines 136-137 - text should be removed
- We removed this.
Brief communication by Strauss and colleagues can make an essential contribution to the topic of bariatric patient care after sleeve gastrectomy but needs to be supplemented to be worthy of publication in Health. The authors should consider the reviewer's comments.
- Thank you for this comment. We have done our best to address reviewer comments.
Reviewer 6 Report
Comments and Suggestions for Authors
The Manuscript entitles as “Evaluating the Impact of Virtual Health Coaching Lifestyle Program on Weight Loss after Sleeve Gastrectomy: A Prospective Study” written well, here are some suggestions that should must be addressed.
· Please rewrite this line “Patients recruited for this 6-month study were classified with a BMI > 30 kg/m2 and had undergone a sleeve gastrectomy at the 90-day post-operative period.” It is grammatically not correct.
· Please add statistical analysis in the abstract section that you have conducted to interpret and analyze the results.
· Give mean values and important results in the abstract section along with their percent increase or decrease.
· Please add a conclusive line in the end of abstract section to summarize the whole abstract.
· It would be much better if keywords are written in alphabetical order.
· Authors should properly explain the sleeve gastrectomy in introduction section.
· Last paragraph of introduction should must present a proper reasoning and rationale of this study.
· In line 79-81 Materials and methodology : Participants; “Patients were recruited for this 6-month study using the following inclusion criteria: were classified with a BMI > 30 kg/m2, had undergone a sleeve gastrectomy at 90-day post-operative period, and had no life-threatening complications related to the surgery or other conditions. Authors should properly indicate the inclusion and exclusion criteria, for instance, how they evaluated the life threatening conditions, what was the main criteria to evaluate the complications of the selected person?? Please mention in details.
· This study was conducted for six months can you please explain how you controlled the patients, how can you be sure that they have followed your schedule? As it was not a controlled study and nor there were any check and balance on the subjects?
· There should be one expert (Psychologist) to ensure and check their routines and other questions related to this study.
· In line 101-108; Patients followed the surgeon's post-operation nutrition plan, which included four phases. 1). 1-3 days of clear liquids, 2) 4-14 days of full liquid, 3). 14-21 days of pureed foods, 4). Soft foods, 5). regular diet. The surgeons recommended 48-64 ounces of water daily and 60 grams of protein, avoiding starches such as bread, chips, pasta, candy, and rice but focusing on complex carbs such as vegetables and legumes. Patients were encouraged to take two chewable vitamins with iron and calcium daily. The surgeon also recommended that patients walk daily and, with time, include strength-based exercise to help with muscle mass maintenance and weight loss? How can you assure they have completely followed the instructions of the surgeon.?
· Results and discussion needs a lot of efforts, there should be proper correlation and reasoning along with the justification and comparison.
· Conclusion should be rewritten with proper summary according to the obtained results along with that authors should also provide their recommendations for community.
· Authors should provide a tabulated or graphical results to highlight the changes and for the easiness of readers.
· Furthermore, They should also add some more parameters to evaluate the complications and other side effects of this treatment.
· In my opinion this research should be controlled and with proper check and balance to obtain the proper results and recommendations for common public.
Comments on the Quality of English Language
Moderate editing of English language required
Author Response
Please rewrite this line “Patients recruited for this 6-month study were classified with a BMI > 30 kg/m2 and had undergone a sleeve gastrectomy at the 90-day post-operative period.” It is grammatically not correct.
Thank you for bringing this point up, we have adjusted that language in the abstract and methods sections.
- Please add statistical analysis in the abstract section that you have conducted to interpret and analyze the results.
Lines 20-21: “repeated measures ANOVA and post-hoc comparison”
- Give mean values and important results in the abstract section along with their percent increase or decrease.
Thank you for mentioning this. Unfortunately, due to the word limit for the abstract based on the journal (200 words), we are unable to add the percentage information.
- Please add a conclusive line in the end of abstract section to summarize the whole abstract.
Lines 25-26: “Further research is needed to assess the long-term effects and cost-effectiveness of such coaching for bariatric surgery patients.”
- Last paragraph of introduction should must present a proper reasoning and rationale of this study.
Thank you for your suggestion; we have added proper reasoning and rationale to the last paragraph of the introduction.
- In line 79-81 Materials and methodology : Participants; “Patients were recruited for this 6-month study using the following inclusion criteria: were classified with a BMI > 30 kg/m2, had undergone a sleeve gastrectomy at 90-day post-operative period, and had no life-threatening complications related to the surgery or other conditions. Authors should properly indicate the inclusion and exclusion criteria, for instance, how they evaluated the life threatening conditions, what was the main criteria to evaluate the complications of the selected person?? Please mention in details.
Line 84-85: “(e.g., chronic conditions, infections, hospital re-admissions).”
- This study was conducted for six months can you please explain how you controlled the patients, how can you be sure that they have followed your schedule? As it was not a controlled study and nor there were any check and balance on the subjects?’
Good point! Language was added to the methods section to address this. Since this is not a randomzied controlled trial, only self-reported information from the patients were reported, and coaches checked in with patients to ensure adherence.
Lines 106-107: “Coaches checked in with patients periodically through the InHealthh ® app to ensure the maintenance of goals.”
- There should be one expert (Psychologist) to ensure and check their routines and other questions related to this study.
All the coaches are national board-certified health coaches, and that information was added to line 93.
- In line 101-108; Patients followed the surgeon's post-operation nutrition plan, which included four phases. 1). 1-3 days of clear liquids, 2) 4-14 days of full liquid, 3). 14-21 days of pureed foods, 4). Soft foods, 5). regular diet. The surgeons recommended 48-64 ounces of water daily and 60 grams of protein, avoiding starches such as bread, chips, pasta, candy, and rice but focusing on complex carbs such as vegetables and legumes. Patients were encouraged to take two chewable vitamins with iron and calcium daily. The surgeon also recommended that patients walk daily and, with time, include strength-based exercise to help with muscle mass maintenance and weight loss? How can you assure they have completely followed the instructions of the surgeon.?
Lines 115-117: To ensure the patients followed surgical plan, constant communication was maintained between coaches and patients, with weekly, biweekly, and monthly check-ins on adherence, healthy lifestyle maintenance, and weight.
- Results and discussion needs a lot of efforts, there should be proper correlation and reasoning along with the justification and comparison.
Thank you for your feedback. We have revised the results and discussion sections to ensure proper correlation, reasoning, and justification, including comparisons where relevant. Additionally, we have added a table to enhance clarity and organization of the data.
- Conclusion should be rewritten with proper summary according to the obtained results along with that authors should also provide their recommendations for community.
Thank you for your feedback. we have ensured that the conclusion provides a comprehensive summary of the obtained results and includes actionable recommendations for the community.
Round 2
Reviewer 1 Report
Comments and Suggestions for Authors
The authors of this publication “Evaluating the Impact of Virtual Health Coaching Lifestyle Program on Weight Loss after Sleeve Gastrectomy: A Prospective Study” has considered most of my remarks made during my first review process. Nevertheless, in my opinion, there are some points to must be fixed additionally:
1. Citation of all the references in the article is not in accordance with all requirements.
2. I suggest including all the tables in the article for more appropriate article reading. Now it is not even clear at what place it will appear.
3. I do not agree that in most research weight is presented in pounds (lbs). Just in United States Pounds are commonly used for general purposes. But International and Scientific Research mostly used Kilograms (kg), because are the standard unit of measurement for weight. Additionally, I would like to note that recalculation of data it is not so difficult.
Comments on the Quality of English Language
The authors of this publication “Evaluating the Impact of Virtual Health Coaching Lifestyle Program on Weight Loss after Sleeve Gastrectomy: A Prospective Study” has considered most of my remarks made during my first review process. Nevertheless, in my opinion, there are some points to must be fixed additionally:
1. Citation of all the references in the article is not in accordance with all requirements.
2. I suggest including all the tables in the article for more appropriate article reading. Now it is not even clear at what place it will appear.
3. I do not agree that in most research weight is presented in pounds (lbs). Just in United States Pounds are commonly used for general purposes. But International and Scientific Research mostly used Kilograms (kg), because are the standard unit of measurement for weight. Additionally, I would like to note that recalculation of data it is not so difficult.
Author Response
The authors of this publication “Evaluating the Impact of Virtual Health Coaching Lifestyle Program on Weight Loss after Sleeve Gastrectomy: A Prospective Study” has considered most of my remarks made during my first review process. Nevertheless, in my opinion, there are some points to must be fixed additionally:
- Citation of all the references in the article is not in accordance with all requirements.
We do not understand this suggestion. Can the reviewer provide more information about how the references in this article are not in accordance with all requirements? We greatly appreciate your feedback in enhancing this publication.
- I suggest including all the tables in the article for more appropriate article reading. Now it is not even clear at what place it will appear.
We thank the review for mentioning this error. We have added the tables and figures into the main document for improved understanding of the results/findings.
- I do not agree that in most research weight is presented in pounds (lbs). Just in United States Pounds are commonly used for general purposes. But International and Scientific Research mostly used Kilograms (kg), because are the standard unit of measurement for weight. Additionally, I would like to note that recalculation of data it is not so difficult.
We thank the reviewer for noting this opinion. Although weight loss is more regularly reported as in lbs., this is more limited to the research conducted and reported from the U.S. As a result, we have adjusted the results to discussing the findings in both lbs., and kg., so weight is reflected for understanding globally across all unit systems used. We did this by reporting the weight in kgs in the table and lbs in the figure, as respective ratio of each other. Again, we thank the review for mentioning this to allow for improved dissemination of research findings.
Reviewer 3 Report
Comments and Suggestions for Authors
Dear authors,
I appreciate your effort in revising the manuscript and for your feedback. This round of revisions has enhanced the quality of the manuscript.
Best,
Comments on the Quality of English Language
Moderate editing of English language required
Author Response
We thank you for the comments.
Reviewer 4 Report
Comments and Suggestions for Authors
Where is the Figure 1?
Author Response
We thank you for the comments. Figure one has been uploaded into the revised manuscript now.
Reviewer 5 Report
Comments and Suggestions for Authors
Dear Authors,
Thank you for including my comments. I believe that in its current form the manuscript is worthy of publication in Healthcare.
Kind Regards
Reviewer
Author Response
We thank you for the comments and appreicate all the great feedback and suggestions to improve our manuscript.
Reviewer 6 Report
Comments and Suggestions for Authors
Authors have revised well and it could be accepted for publication
Comments on the Quality of English LanguageMinor editing of English language required
Author Response

(The authors gave the same response as above.)
